# Is the A_-1_ Pigment in Photosystem I Part of P700? A (P700^+^–P700) FTIR Difference Spectroscopy Study of A_-1_ Mutants

**DOI:** 10.3390/ijms25094839

**Published:** 2024-04-29

**Authors:** Julia S. Kirpich, Lujun Luo, Michael R. Nelson, Neva Agarwala, Wu Xu, Gary Hastings

**Affiliations:** 1Department of Physics and Astronomy, Georgia State University, Atlanta, GA 30303, USA; 2Department of Chemistry, University of Louisiana at Lafayette, Lafayette, LA 70504, USA

**Keywords:** photosystem I, P700, A_-1_, chlorophyll *a*, FTIR difference spectroscopy, mutant

## Abstract

The involvement of the second pair of chlorophylls, termed A_-1A_ and A_-1B_, in light-induced electron transfer in photosystem I (PSI) is currently debated. Asparagines at PsaA600 and PsaB582 are involved in coordinating the A_-1B_ and A_-1A_ pigments, respectively. Here we have mutated these asparagine residues to methionine in two single mutants and a double mutant in PSI from *Synechocystis* sp. PCC 6803, which we term NA600M, NB582M, and NA600M/NB582M mutants. (P700^+^–P700) FTIR difference spectra (DS) at 293 K were obtained for the wild-type and the three mutant PSI samples. The wild-type and mutant FTIR DS differ considerably. This difference indicates that the observed changes in the (P700^+^–P700) FTIR DS cannot be due to only the P_A_ and P_B_ pigments of P700. Comparison of the wild-type and mutant FTIR DS allows the assignment of different features to both A_-1_ pigments in the FTIR DS for wild-type PSI and assesses how these features shift upon cation formation and upon mutation. While the exact role the A_-1_ pigments play in the species we call P700 is unclear, we demonstrate that the vibrational modes of the A_-1A_ and A_-1B_ pigments are modified upon P700^+^ formation. Previously, we showed that the A_-1_ pigments contribute to P700 in green algae. In this manuscript, we demonstrate that this is also the case in cyanobacterial PSI. The nature of the mutation-induced changes in algal and cyanobacterial PSI is similar and can be considered within the same framework, suggesting a universality in the nature of P700 in different photosynthetic organisms.

## 1. Introduction

Photosystem I (PSI) is a membrane-spanning protein complex found in all photosynthetic oxygen-evolving organisms. PSI uses light energy to transfer electrons across the thylakoid membrane, resulting in reducing the power that can be used in the Calvin-Benson cycle [1]. To study the nature of the cofactors involved in electron transfer (ET) in PSI, isolated photosynthetic protein complexes are mostly studied. In these isolated complexes, light induces ET, resulting in a radical pair state that eventually recombines in hundreds of milliseconds to seconds, depending on the presence or absence of artificial electron donors and acceptors [2,3,4].

A broadly accepted idea is that PSI light excites a special pair of strongly coupled chlorophyll molecules that are together called P700 [5]. These “special” chlorophyll molecules are termed P_A_ and P_B_. The subscripts refer to which protein (PsaA or PsaB) the pigments are ligated to. Light excitation of P700 (without specifying exactly what P700 is) leads to an ET process along one of the two nearly symmetrical ET branches, called the A- and B-branches. The nature of the primary electron acceptor, and the true nature of the primary electron donor for that matter, is still an unresolved and debated subject, which is one reason for the current study. Currently, there are three models that define the nature and role of the cofactors involved in ET in PSI. In the first model, the A_-1_ chlorophylls act as accessory chlorophylls without a particular role as an intermediate in ET [4]. In the second model, the A_-1_ chlorophylls act as primary electron donors [6,7,8,9], and in the third model, the A_-1_ pigments are part of a collective multimeric pigment species [10,11]. Thus, the different models hinge on the properties of the A_-1_ pigment. What is known with some certainty is that following light excitation, the P700^+^A_1_^–^ state is formed within tens of picoseconds [12,13], with the cation state residing on the P_A_ and/or P_B_ pigments [5,14].

The most commonly accepted idea is that the P_A_ or P_B_ pigments of P700 constitute the primary electron donor, and following light excitation of this donor, an electron is transferred to the A_0_ chlorophyll [4]. From high-resolution PSI crystal structures [15], it was established that there were two chlorophyll molecules on the A and B branches, between the P_A_ and P_B_ pigments of P700 and A_0_. These were mostly considered to be accessory chlorophylls (Chls), in analogy to a similar situation observed in purple bacterial reaction centers (RCs), where an accessory bacteriochlorophyll pigment is located between the primary donor and acceptor species. The structure and numbering scheme of Chl *a* are shown in Figure 1.

Unlike the situation in purple bacterial RCs, where only six pigments are present, isolated PSI complexes contain ~100 Chl molecules that act as antenna pigments, funneling excitation energy to the primary electron donor (P700). This funneling process takes ~50 ps [12], on a similar timescale to the primary ET processes. In addition, the large number of pigments makes it near impossible to directly excite the P700 species and hence follow the ET events without interference. Since the P700^+^A_1_^–^ state has a lifetime in the nanosecond regime [4,16,17], it is easily probed, as all excited antenna pigments have decayed. Given that it is near impossible to *directly* probe P700^+^A_-1_^–^ and P700^+^A_0_^–^ formation and decay, alternative and more indirect methods are required.

It has been established that both A- and B-branches participate in ET in PSI. The degree of directionality is species-dependent, with an ~80/20 bias toward the A-branch in cyanobacterial PSI [18,19,20,21] and closer to 50/50 in green algal PSI [22]. However, there is no definitive understanding if the two branches are in equilibrium at one or a few steps of ET or if they are independent, and an electron follows only one branch once it starts its journey. There are two models regarding current hypotheses on branch interactions: the “donor-side equilibrium model” and the “branch competition model” [23]. In the former, the two branches act independently of each other, while in the latter, the branches are at equilibrium and an electron is passed from one branch to another under appropriate conditions.

ET in PSI is roughly described as a process where electrons “hop” from one pigment to another. Amino acid residues near the pigments can directly impact this ET process. For example, Figure 2 shows the organization of the A_-1_ pigments in their binding sites, indicating that asparagine (Asn or N) residues NA600 and NB582 are hydrogen bonded (H-bonded) to a water molecule that in turn is ligated to the A_-1B_ or A_-1A_ pigments, respectively. Figure 2 shows structures in PSI from *Synechocystis* sp. PCC 6803 (*S6803*). The distances from the Asn carbonyl oxygen atom to the water oxygen atom and to the carbonyl oxygen atoms of the closest Chls are also shown in Figure 2 and are listed in Table 1. The Chl *a* numbering scheme is indicated in Figure 1 and is used in Table 1. The structures corresponding to those in Figure 2 for PSI from *Chlamydomonas reinhardtii* (*C. reinhardtii*) (6JO5) [24] are shown in Appendix A (distances indicated in Appendix A). The structures for PSI from *S6803* and *C. reinhardtii* are mostly similar (with distance differences well below 1 Å). One outlier is the carbonyl (C=O) oxygen atom of NA600 in *S6803*, which is 1.6 Å farther from the 13^3^-ester C=O oxygen atom of P_A_ than that found in *C. reinhardtii* (see Figure 1 for Chl *a* numbering). This difference arises because the 13^3^-ester C=O of P_A_ points away from the A_-1B_ pigment in PSI from *S6803* (Figure 2A) but points toward A_-1B_ in PSI from *C. reinhardtii* (Appendix A). This species-specific structural difference is not found for the A_-1A_ pigment on the B-branch.

In this manuscript, we examine NA600M, NB582M, and NA600M/NB582M A_-1_ mutants in PSI from *S6803*. These same mutants have previously been the subject of ultrafast visible, EPR, and time-resolved fluorescence spectroscopy studies [10,11,18,26]. According to Badshah et al. [18], Cherepanov et al. [10], and Martin [26], the stable charge separation yield (yield of the “long-lived” P700^+^A_1_^–^ state) is the same as in WT, in both the NA600M and NB582M single mutants. In the NA600M/NB582M double mutant, however, the yield of P700^+^A_1_^–^ decreases [26]. In spite of the above, however, in all three mutants there is a decrease in the yield of primary charge separation (P700^+^A_-1_^–^) compared to the WT. Specifically, A-branch mutants show a more pronounced reduction in primary charge separation yield than B-branch mutants, with the double mutant exhibiting the most significant suppression. While stable charge separation remains unchanged and primary charge separation is reduced in A_-1_ mutants, the exact impact of Asn-to-Met mutations on P700 in both neutral and cation states remains unclear. Various potential impacts on A_-1_ have been discussed, including hydrogen bonding modification [10,11], water ligand loss [26], and possible hydrogen bond formation with the introduced amino acids [11]. To gain a clearer understanding of the mutation-induced impact on P700, we have been using FTIR difference spectroscopy (DS).

FTIR DS is sensitive to atomic-scale structural changes in pigment molecular groups and to small changes in H-bonding. The frequencies of the C=O bands of Chl molecules depend on the presence and number of hydrogen bonds, as well as the polarity of the environment [27]. This makes FTIR DS an ideal probe for mutations that impact the A_-1_ pigment. In this manuscript, we have obtained light-induced (P700^+^–P700) FTIR DS for WT PSI from *S6803* and on PSI mutants where the Asn residues near the A_-1_ pigments have been changed to methionine (NA600M, NB582M, and NA600M/NB582M). These DS inform us on how these residues impact the pigments and how or if these pigments are involved in ET in the WT and mutant systems.

## 2. Results and Discussion

The (P700^+^–P700) FTIR DS for WT and mutant PSI samples at RT (293 K) are shown in Figure 3. By subtracting (P700^+^–P700) FTIR DS for WT PSI from the corresponding mutant DS, “mutant minus WT” double difference spectra (DDS) are constructed and are also shown in Figure 3. The DDS allows a clearer visualization of mutation-induced alterations in the DS. Bands in the DDS should mostly correspond to molecular groups near the site of mutation that are directly impacted by the mutation or to the mutated residues (Asn and Met) themselves. More distant molecular groups that are impacted electrochromically in different ways in the WT and mutant may also exhibit features in the DDS.

Spectral alterations (compared to WT) in the NA600M mutant are more extensive than those in the NB582M mutant. FTIR DS for the double mutant are considerably modified compared to WT, so much so that these DS are almost unrecognizable as (P700^+^–P700) FTIR DS. The species that we refer to as P700 is very heavily disrupted in the double mutant.

### 2.1. Infrared Band Assignments

Many of the bands in the WT FTIR DS in Figure 3 have been assigned to the 13^1^-keto and 13^3^-ester C=O groups of the P_A_ and P_B_ pigments [28,29]. These assignments were made based on the PSI crystal structure of *Thermosynechococcus elongatus* [15], which indicated that P700 appeared to be mostly a heterodimer consisting of the P_A_ and P_B_ pigments [28]. It has not been widely recognized that these assignments are incomplete, with many of the bands in the FTIR DS displaying shoulders that indicate several molecular components may contribute to the bands in the FTIR DS.

For example, the negative 1697 cm^−1^ band in the WT FTIR DS has been assigned to the 13^1^-keto C=O group of the P_B_ pigment of P700 [14,30]. A first indication that multiple species could contribute to the 1697 cm^–1^ band came from a FTIR study of a *C. reinhardtii* mutant where the histidine residue that ligates the P_A_ pigment is changed to Ser [31]. In this mutant, the negative band at 1697 cm^−1^ splits into two distinct bands. This is a difficult observation to rationalize based on the hypothesis that the whole 1697 cm^−1^ band is due to the 13^1^ keto C=O group of P_B_ (the mutation is at P_A_). This observed mutation-induced splitting of the 1697 cm^−1^ band of WT gave a hint that molecular groups of pigments other than P_B_ could contribute to bands in the WT FTIR DS. Following on from this previous work [31], the FTIR DS presented here also indicate a considerable alteration of the 1697 cm^−1^ band of WT upon mutation (Figure 3). Such large-scale alterations in the 1697 cm^−1^ band were also noted in a recent FTIR DS study on other A_-1_ mutants in PSI from *C. reinhardtii* [32].

The mutation-induced spectral alterations in the 1697 cm^−1^ band shown in Figure 3 are interesting given that the site of the mutation on either branch is distant (~20 Å) from the P_B_ 13^1^-keto carbonyl oxygen atom (Figure 2). From these structural data, it is reasonable to conclude that the 13^1^-keto C=O group of the A_-1_ pigments could contribute to changes near 1697 cm^−1^ in WT PSI FTIR DS. However, the Asn-to-Met mutation is not expected to greatly alter the C=O groups of the A_-1_ pigments. Despite this expectation, considerable alterations in FTIR DS are observed upon mutation, suggesting that the A_-1_ pigments do contribute considerably to both WT and mutant FTIR DS.

### 2.2. Bands Associated with A_-1B_

In the NA600M–WT DDS in Figure 3A (*black*), an intense quadruple feature is observed at 1710(−)/1697(+)/1691(+)/1683(−) cm^−1^. The positive peak at 1691 cm^−1^ is more of a shoulder than a clear peak. There are several hypotheses as to the origin of this quadruple feature:

#### 2.2.1. A C=O Group of Asn

Since Asn is changed to Met, one hypothesis is that there may be changes in the DS due to the loss of the C=O group of Asn that is present in WT PSI. However, this change might result only in the appearance of a first derivative-like feature in the DDS. The presence of such a first derivative-like feature would also imply that the Asn C=O mode shifts upon cation formation in WT PSI. Even if the Asn C=O mode is perturbed upon cation formation, it is still unlikely that this would give rise to an intense band in the FTIR DS because bands associated with amino acid side chains are usually very weak in (P700^+^–P700) FTIR DS [28]. That is, most mutation-induced shifting of bands in FTIR DS is due to molecular groups of pigments that are modified because they interact with the amino acid that is changed.

If part of the quadruple feature is due to the C=O group of NA600, then one might also expect to observe an NH_2_ bending mode of NA600 near 1610 cm^−1^ [33]. Such a feature is not observed. The above considerations allow us to rule out a contribution from Asn C=O modes to the quadruple feature in the DDS in Figure 3A.

#### 2.2.2. Amide (Protein) Vibrations

The quadruple feature in Figure 3A is sufficiently high in frequency that we can rule out the idea that part of the feature may be due to amide vibrations. Amide I modes of *a*-helical protein segments occur near 1655 cm^−1^. *β*-sheet segments can give rise to vibrational modes up to ~1690 cm^−1^ [34], but no such *β*-sheet structural elements are observed in PSI X-ray structures.

#### 2.2.3. The 13^1^-Keto C=O Group of A_-1B_

The quadruple feature in the DDS in Figure 3A could be due to the 13^1^-keto C=O group of a Chl *a* molecule that is modified upon mutation of Asn to Met. Given the observed quadruple feature, this hypothesis implies that this 13^1^-keto C=O group is impacted by cation formation in *both* WT and mutant PSI.

A similar (but somewhat shifted) quadruple feature is also observed in the NB582M–WT DDS in Figure 3B. In fact, such a feature is also observed in corresponding Asn to Lys or Asp mutants in PSI from *C. reinhardtii* (Table 2 and Table 3) [32]. Since a similar quadruple feature is observed in DDS with corresponding mutations on either branch, we conclude that mutation-induced modifications impact both branches similarly. This then suggests that the features are not due to the 13^1^-keto C=O groups of the P_A_ or P_B_ pigments, since the 13^1^-keto C=O groups of P_A_ and P_B_ are at very different frequencies (P_B_/P_A_ bands at~1697/1639 cm^−1^, respectively [28,29]). The quadruple features in Figure 3A,B are therefore unlikely to be associated with the P_A_ and P_B_ pigments, and we are led to the conclusion that in WT PSI FTIR DS there is a band that is due to a 13^1^-keto C=O group of a Chl *a* species. This species is probably the A_-1_ pigment, based purely on a proximity argument (Figure 2).

Since the DDS for the NA600M mutant displays a positive feature at 1697 cm^−1^ (Figure 3A), it suggests a 13^1^-keto C=O group could absorb near 1697 cm^−1^ in WT PSI. The DDS indicate this feature could upshift ~13 cm^−1^ to 1710 cm^−1^ or downshift 14 cm^−1^ to 1683 cm^−1^ upon cation formation. A downshift upon cation formation suggests that the charge in the P700^+^ state is not distributed over the A_-1_ pigment (because in that case an upshift would be expected). There are two possible scenarios in which a downshift might be expected. Firstly, it is possible that the A_-1_ pigment is electronically coupled to the P_A_ and P_B_ pigments in the ground state, but this coupling is lost upon cation formation. Secondly, it is possible that there is an electrochromic effect (from the charge on P_A_ and/or P_B_) on the A_-1_ pigment molecular groups. In this case, the A_-1_ pigment need not necessarily be electronically coupled to the P_A_ and/or P_B_ pigments. In both scenarios a 14 cm^–1^ downshift *upon cation formation* is comparable to an *anion-induced downshift* of chlorophyll in solution [30,35], and is therefore unlikely. More likely is that the 13^1^-keto C=O group in WT, at 1697 cm^−1^, upshifts to 1710 cm^−1^.

Thus far, we have assigned the 1697(+)/1710(−) cm^−1^ part of the quadruple feature in the DDS in Figure 3A to the cation-induced upshift of a band of the A_-1B_ pigment. The other part of the quadruple feature is at ~1691(+)1683(−) cm^−1^. We propose that *in the NA600M mutant*, the band at ~1683 cm^−1^ upshifts ~8 cm^−1^ to ~1691 cm^−1^ upon cation formation. Furthermore, this also indicates a 14 cm^−1^ mutation-induced downshift (1697 cm^−1^ in WT downshifts to 1683 cm^−1^ in the mutant (Table 3)). Such a large shift likely indicates the 1683 cm^−1^ band in the DDS (Figure 3A) is due to the 13^1^-keto C=O of A_-1B_ (Table 2), since the Asn residue at A600 is close (~10.1 Å) to the A_-1B_ pigment (Figure 2A, Table 1).

When AsnA600 and AsnB582 are mutated to Met, it is likely that the H-bond to the nearby water molecule will be eliminated and the ligand to the A_-1_ pigments will be lost, potentially leading to some structural alteration of the A_-1_ pigments. The impact such a structural alteration will have on the 13^1^-keto and 13^3^-ester C=O groups of the A_-1_, P_A_, and P_B_ pigments is difficult to assess.

In summary, the NA600M–WT DDS in Figure 3A indicates a 13^1^-keto C=O band of the A_-1B_ pigment at 1697 cm^−1^ in the WT PSI that upshifts 13 cm^−1^ to 1710 cm^−1^ upon cation formation (Table 2). In the mutant, the 13^1^-keto C=O of the A_-1B_ pigment is at 1683 cm^−1^ and upshifts 8 cm^−1^ to 1691 cm^−1^ upon cation formation (Table 2). This further implies a 14 cm^−1^ mutation-induced downshift of the A_-1B_ C=O mode (Table 3). The magnitudes of these shifts are large and suggest that A_-1_ vibrational modes are significantly affected by P700^+^ cation formation. This might suggest that electronic charge is distributed over the A_-1_ pigments and that P700 is a true multimeric coupled pigment species.

### 2.3. Bands Associated with A_-1A_

(P700^+^–P700) FTIR DDS for the NA600M mutant differs somewhat from that for the NB582M mutant (Figure 3A,B). Hence, the two branches are not entirely symmetric. NB582M mutant FTIR DDS display a second-derivative-like feature at 1699(−)/1692(+)/1679(−) cm^−1^. This lack of observation of a clear quadruple feature is the consequence of the different overlap of four different bands due to WT and mutant in the neutral and cation states.

The positive band at 1692 cm^−1^ in the DDS in Figure 3B suggests a 13^1^-keto C=O band of a pigment in WT PSI in the neutral state. It is unlikely that this band is due to P_A_ since the 13^1^-keto C=O group of P_A_ is distant from the site of mutation (19.9 Å, Table 1, Figure 2B). Hence, based on proximity, we suggest that at least part of the positive 1692 cm^−1^ feature in the DDS in Figure 3B is due to the 13^1^-keto C=O mode of A_-1A_.

The second derivative-like feature at 1699(−)/1692(+)/1679(−) cm^−1^ in the DDS in Figure 3B suggests that the 13^1^-keto C=O mode of A_-1A_ is shifted both upon cation formation and by mutation. Hence, bands associated with A_-1A_ are present in the WT FTIR DS. Therefore, for WT PSI, the A_-1A_ species displays a band at 1692(−) cm^−1^ that upshifts 7 cm^−1^ to 1699(+) cm^−1^ upon cation formation (Table 2). In the mutants, the corresponding bands are at 1679(−)/1692(+) cm^−1^ and are thus downshifted 13/7 cm^−1^ upon mutation (Table 3). While the 1692(−)/1699(+) cm^−1^ difference band of A_-1A_ is not resolved in the WT FTIR DS, comparison with mutant FTIR DS helps to uncover it.

The upshift in the 13^1^-keto C=O group of A_-1A_ upon P700^+^ formation is 7 cm^−1^ (Table 2). This could suggest either that A_-1A_ is affected electrostatically by the positive charge on P_A_ and/or P_B_ or that the positive charge is distributed to some degree over the A_-1A_ pigment, in which case the A_-1A_ pigment is an integral part of P700.

As suggested above for the NA600 mutant, mutation of NA582 to Met will likely lead to loss of the water molecule that ligates A_-1A_ (based on changing Asn to Met using Chimera software 1.17.3, it seems unlikely that the water molecule will remain H-bonding to the introduced Met). This may cause a structural alteration of A_-1A_. How this mutation-induced alteration may impact the molecular groups of A_-1A_, P_A_, and P_B_ (both structurally and electrostatically) is unclear at present.

### 2.4. 13^3^-Ester C=O Bands of A_-1_

In FTIR DS obtained for *C. reinhardtii* A_-1_ mutants, a band is observed at 1734(−)/1728(+) cm^−1^ in [NB587D–WT] FTIR DDS. No such band is observed in the [NB587K–WT] FTIR DDS [32]. The C=O mode of aspartic acid (Asp, D) was suggested to explain the presence of this band in [NB587D–WT] FTIR DDS. However, it is at a considerably lower frequency than might be expected for a carboxylic acid C=O group. Alternatively, the 1734(−)/1728(+) cm^−1^ feature could be assigned to the 13^3^-ester C=O group of A_-1A_. However, the absence of such a feature in [NB587K–WT] DDS pointed potentially to structural differences in the NB587D and NB587K mutants from *C. reinhardtii*.

In the Asn-to-Met mutants for *S6803* discussed here, no bands are observed in the DDS above ~1730 cm^−1^. Hence, we do not observe features that can be easily assigned to the 13^3^-ester C=O groups of the A_-1_ pigments. It may be the case that such changes lead to only very weak features in a DDS that cannot be resolved. We also point out that the spectra presented here have been normalized so that the difference band near 1754(+)/1748(−) cm^−1^ has similar intensity. This normalization seems entirely appropriate based on the FTIR DS in Appendix A. This normalization, however, might have the effect of minimizing features near ~1750 cm^−1^ in the DDS. The present data suggest that mutation induced changes in frequency of the 13^3^-ester C=O modes of the A_-1_ pigments contribute negligibly to the DDS.

The summary of band assignments is as follows:In WT PSI, the 13^1^-keto C=O group of A_-1B_ is found at 1697 cm^−1^ and upshifts 13 cm^−1^ to 1710 cm^−1^ upon cation formation. When Asn at A600 is changed to Met, the 1697(−)/1710(+) cm^−1^ difference band downshifts 16/19 cm^−1^ to 1683(−)/1691(+) cm^−1^, respectively.In WT PSI, the 13^1^-keto C=O group of A_-1A_ is found at 1692 cm^−1^ and upshifts by 7 cm^−1^ to 1699 cm^−1^ upon cation formation. When Asn at B582 is changed to Met, the 1692(−)/1699(+) cm^−1^ difference band downshifts 13/7 cm^−1^ to 1679(−)/1692(+) cm^−1^.

### 2.5. Additivity of Mutation-Induced Spectral Changes

The DDS for the double mutant in Figure 3C indicates that the features are more intense compared to the DDS for the single mutants. In addition, the DDS for the double mutant is broader, displaying a broad positive peak near 1694 cm^−1^ and negative peaks at 1710 and 1680 cm^−1^. These two negative peaks are similar to those found in each of the DDS for the single mutants and therefore suggest that the double mutant DDS can be considered a combination of the two single mutant DDS. The amplitude and width of the positive peak near 1694 cm^−1^ in the DDS for the double mutant also suggest this possibility. To conclude, the mutation-induced spectral alterations appear to be approximately additive. To better visualize this correlation, Figure 4A shows the DDS for the two single mutants and the double mutant overlaid. Figure 4B shows a DDS obtained by adding a linear combination of the two DDS for the single mutants and comparing that to the double mutant DDS.

To produce the composite spectrum in Figure 4B, the (NA600M–WT) DDS is scaled by 3.0, while the (NB582M–WT) DDS is scaled by 0.75. Although this 4:1 linear combination was obtained empirically, it suggests a larger impact of A-branch mutations, which in turn might suggest a greater degree of ET down the A-branch compared to the B-branch. This conclusion is in line with a previous report on A_-1_ mutants [18]. The additivity of the single mutant DDS supports their independent contributions to the negative 1699 and 1710 cm^−1^ bands in each of the DDS. This finding is relevant to the debate between the two main hypotheses about A and B branch interactions, namely, the donor-side equilibrium model and the branch competition model. The observed additivity may favor the donor-side equilibrium model, where the two branches function independently. However, it is important to note that this additivity was only observed in the range of 1760–1670 cm^−1^ and not between 1670–1625 cm^−1^ (Figure 4B). Badshah et al. [18] reported that ET yield on the branch opposite to the mutated one did not increase to compensate for a decrease in yield on the mutated branch, which agrees with the idea that the branches function independently. However, Cherepanov et al. [11] and Martin et al. [26] support a branch competition model in which the branches compensate for deficiencies in ET introduced on the opposite branch. Clearly, this debate requires further experiments to help resolve it. The fact that ET is not completely blocked in the double mutant indicates that the mutations do not completely hinder A_–1_ participation in ET, either as part of the electron donor or as an acceptor.

## 3. Materials and Methods

### 3.1. PSI Particle Preparation

The recipient strain pWX3 of *S6803* was modified for site-directed mutagenesis of the psaA and psaB genes by deleting a portion of psaA and the entire psaB gene, as previously described in [36]. The pIBC plasmid was created to generate a site-specific mutation in the psaA gene (NA600M) by cloning a DNA fragment containing most of psaA, psaB, and a 760 bp downstream region of the psaB gene into the pBluescript II KS vector and inserting a chloramphenicol resistant cassette gene after the 3′ terminator of the psaB gene. Similarly, the plasmid pBC was constructed to generate a site-specific mutation in the psaB gene (NB582M) by cloning a 1588-bp region of the psaB 3′ region and a 760-bp downstream region into the pBluescript II KS vector and inserting a chloramphenicol resistance gene at the *Eco*RI site just downstream of psaB. The recipient strain pCRTΔB was used for site-directed mutagenesis of the psaB gene using PCR mutagenesis with the QuikChange site-directed mutagenesis kit (Stratagene, La Jolla, CA, USA). The mutated plasmid constructs were verified by sequencing and transformed into *S6803* recipient strains, with segregation of the transformants confirmed through screening for chloramphenicol resistance. Full segregation of the desired NA600M and NB582M variants was confirmed by PCR and DNA sequencing. PSI complexes from the WT and the NA600M and NB582M variants were isolated, as described in [36]. PSI samples were stored at −81 °C until use.

### 3.2. FTIR Sample Preparation

For FTIR measurements, PSI samples at ~0.2 mg/mL chlorophyll were dissolved in 50 mM Tris with 0.04% *β*-DM (*w*/*v*) and ultra-centrifuged at 400,000× *g* in a TLA-100 rotor for 2–3.5 h at 4 °C. The obtained pellet was transferred onto a CaF_2_ window, and ~0.5 µL drops of 20 mM sodium ascorbate and 15 µM phenazine methosulfate (PMS) were added to the pellet. After a period (~5 min) of drying, the pellet was squeezed between two CaF_2_ windows. The sample absorption was adjusted to ~1.0 OD near 1656 cm^−1^. Three to five independently prepared samples for each mutant were studied. This allowed an estimate of experimental variability.

### 3.3. FTIR Spectroscopy

Photo-accumulation FTIR DS measurements were conducted using a Vertex 80 FTIR spectrometer (Bruker Optics, Billerica, MA, USA). The sample was illuminated using a 15 mW HeNe laser adjusted to a spot size of ~1 cm at the sample. Single beam absorbance spectra obtained for samples under illumination were ratioed directly against absorbance spectra collected for samples in the dark, allowing the construction of what we refer to as [P700^+^–P700] FTIR DS. In these FTIR DS, the positive and negative bands are due to P700^+^ and P700, respectively. For each mutant, the light/dark cycle was repeated ~100 times, and the FTIR DS were averaged. Spectra were collected in the 7000–1250 cm^−1^ region. Only the 1780–1600 cm^−1^ region is considered in this manuscript (see Appendix A for FTIR DS over a broader spectral region).

## 4. Conclusions

This study used FTIR DS to investigate changes to P700 in mutants where the ligands to the A_-1_ pigments are altered. Bands in the FTIR DS that had previously been assigned to the P_B_ pigment of P700/P700^+^ are clearly modified. The conclusion is that the A_-1_ pigments also contribute to these bands that had previously been assigned to P_B_. The accepted sets of assignments of bands in (P700^+^–P700) FTIR DS are therefore shown to be incomplete.

Here we developed a model to explain the FTIR DDS, where bands of both A_-1A_ and A_-1B_ are shown to upshift upon cation formation, and both the neutral and cation bands of WT are shown to downshift upon mutation. A similar model was developed previously for A_-1_ mutants from *C. reinhardtii*, demonstrating a universality over species in the nature of the A_-1_ pigment via its spectroscopic properties. We tentatively suggest that the A_-1_ pigments are coupled electronically with the P_A_ and P_B_ pigments, producing the multimeric species P700.

## Figures and Tables

**Figure 1 ijms-25-04839-f001:**
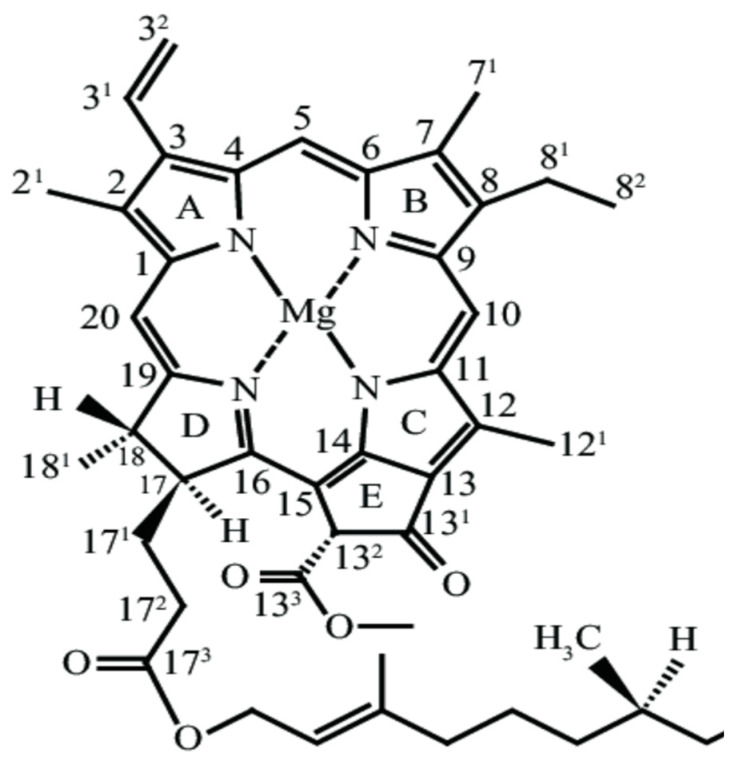
Chemical structure of Chl *a* with the International Union of Pure and Applied Chemistry (IUPAC) numbering scheme.

**Figure 2 ijms-25-04839-f002:**
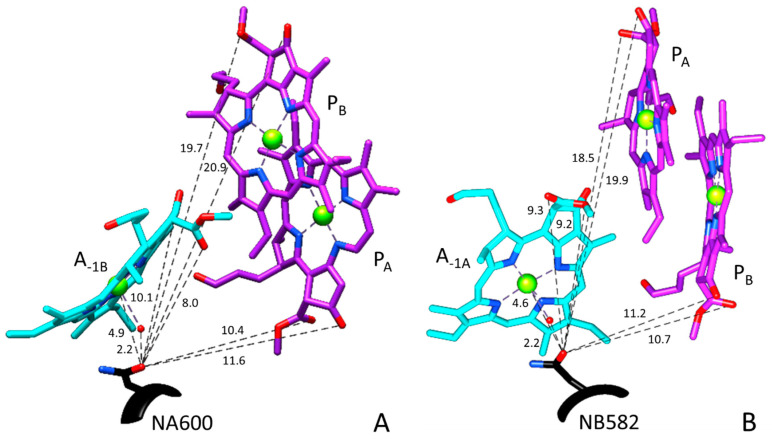
Structural organization of the A_-1A_, A_-1B_, P_A_, and P_B_ pigments (**A**) and A_-1B_, P_A_, and P_B_ pigments (**B**). P_A_ and P_B_ are shown in purple and A_-1A_ in cyan. Structure obtained using UCSF Chimera software 1.17.3 utilizing the PSI crystal structure from *S6803* (PDB file 5OY0 [25]) at 2.5 Å resolution. The backbone and side chain of Asn at A600 and B582 are also shown (*black*). Distances from the Asn carbonyl oxygen atoms (*dashed lines*) are shown (in Å). These distances are listed in Table 1.

**Figure 3 ijms-25-04839-f003:**
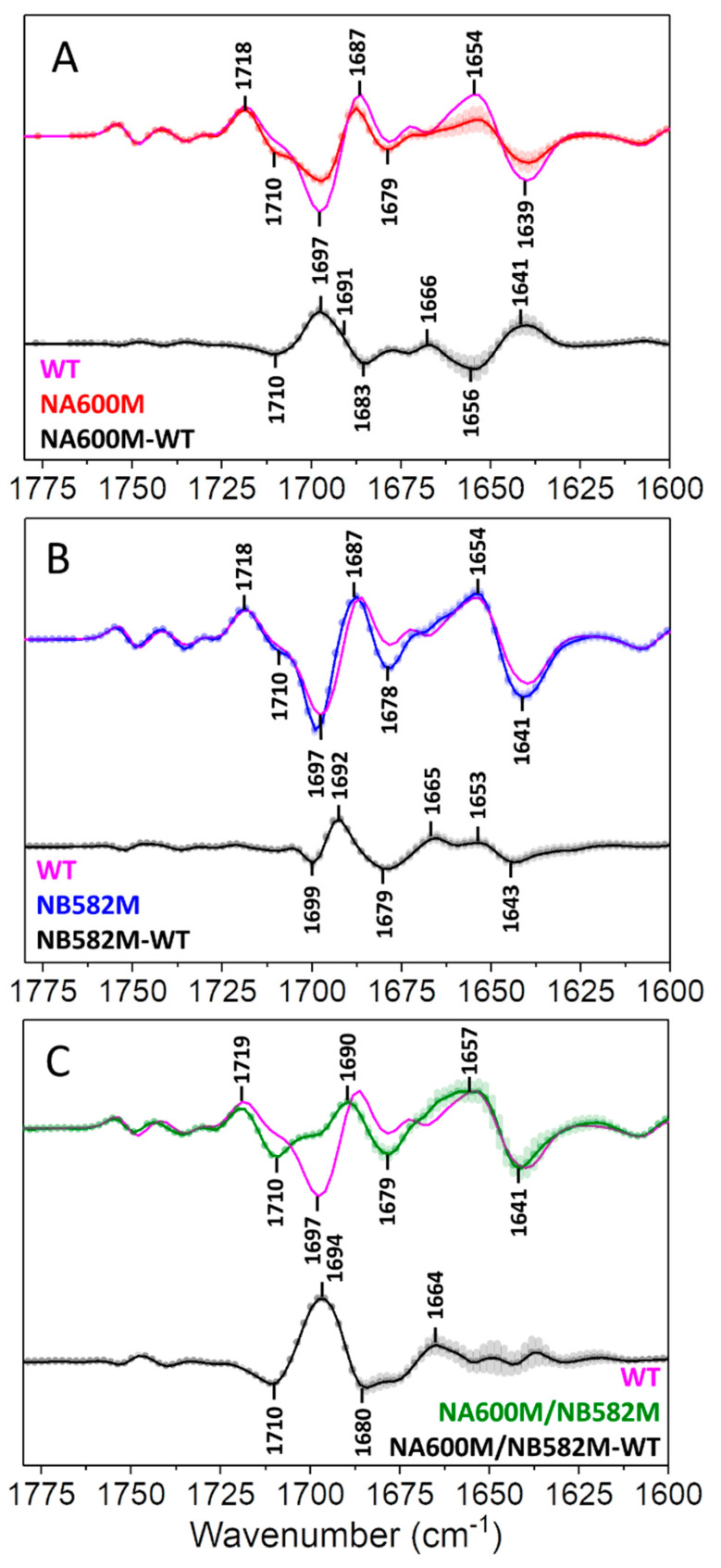
The (P700^+^–P700) FTIR DS in the 1780–1600 cm^−1^ region for the (**A**) NA600M (*red*), (**B**) NB582M (*blue*), and (**C**) NA600M/NB582M (*green*) mutants at RT (293 K). The corresponding DS for WT PSI is also shown (*magenta*) in each of the three panels. Spectra are the average of at least three measurements. The WT and mutant spectra were scaled so the 1748(−)/1754(+) cm^−1^ difference bands were similar. This scaling minimizes the differences in the spectra over the entire 1800–1200 cm^−1^ region (see Appendix A). DDS (*black*) were constructed by subtracting the WT from the mutant DS. Shaded areas indicate the standard error in repeated experiments. The propagated standard error is shown in the DDS.

**Figure 4 ijms-25-04839-f004:**
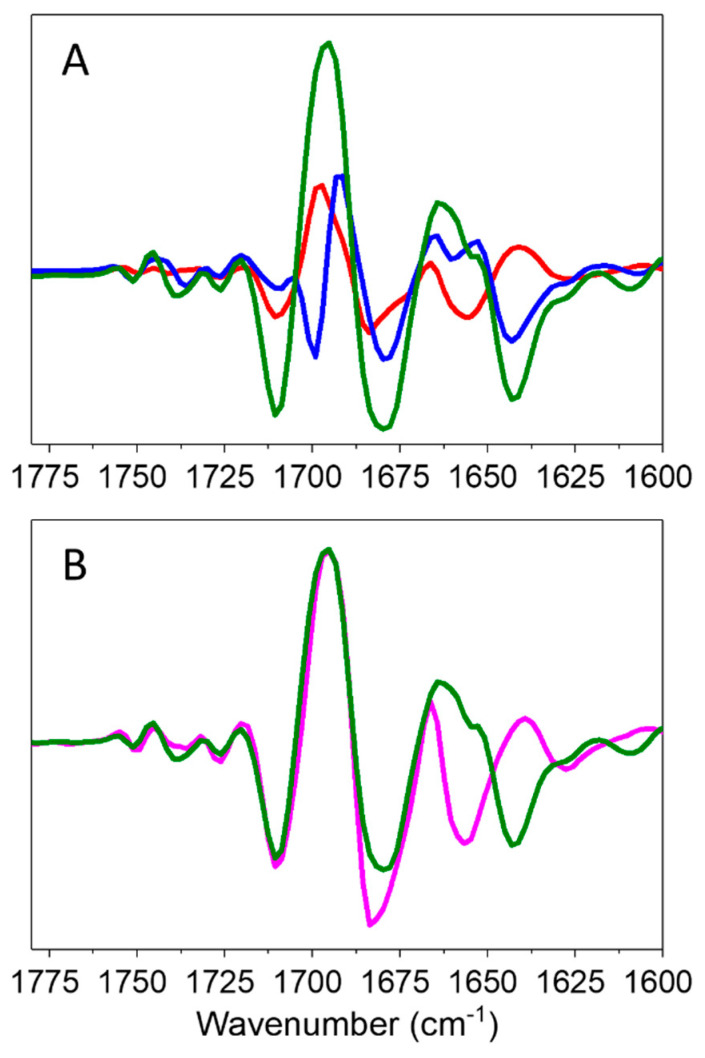
(**A**) Comparison of [NA600M−WT] (*red*), [NB582M−WT] (*blue*), and [NA600M/NB582M–WT] double mutant (*green*) FTIR DDS that are presented in Figure 3A–C. (**B**) DDS obtained by taking a linear combination of the two single mutant DDS (3 × (NA600M–WT) + 0.75 × (NB582M–WT), *magenta*) compared to the double mutant DDS (*green*).

**Table 1 ijms-25-04839-t001:** Distances (Å) between the C=O oxygen atom of the Asn residues on the B and A branches and various atoms of the P_A_ and P_B_ pigments. Distances are from the PSI structure in *S6803* (PDB file 5OY0).

	NB582 C=O	NA600 C=O
H_2_O oxygen atom	2.2	2.2
Mg atom	4.6	4.9
13^1^-keto C=O oxygen atom of P_A_	19.9	11.6
13^3^-ester C=O oxygen atom of P_A_	18.5	10.4
13^1^-keto C=O oxygen atom of P_B_	11.2	20.9
13^3^-ester C=O oxygen atom of P_B_	10.7	19.7
13^1^-keto C=O oxygen atom of A_-1A_	**9.4**	15.2 (not in Figure 1)
13^3^-ester C=O oxygen atom of A_-1A_	**9.3**	18.7 (not in Figure 1)
13^1^-keto C=O oxygen atom of A_-1B_	14.3 (not in Figure 1)	**10.1**
13^3^-ester C=O oxygen atom of A_-1B_	17.4 (not in Figure 1)	**8.0**

**Table 2 ijms-25-04839-t002:** Band assignment in (P700^+^–P700) FTIR DS obtained using WT and A_-1_ mutant PSI from *S6803* and *C. reinhardtii* at 293–298 K. Frequency differences upon cation formation are indicated by the symbol Δ and are in cm^−1^.

*S6803*						
	WT	Δ	N-to-M Mutant		Δ	
13^1^-keto C=O A_-1B_	1697(−)/1710(+)	13	1683(−)/1691(+)		8	
13^1^-keto C=O A_-1A_	1692(−)/1699(+)	7	1679(−)/1692(+)		13	
** *C. reinhardtii* **						
	**WT**	**Δ**	**N-to-K Mutant**	**Δ**	**N-to-D Mutant**	**Δ**
13^1^-keto C=O A_-1B_	1692(−)/1702(+) 1691(−)/1700(+)	10 9	1674(−)/1688(+)	14	1673(−)/1688(+)	15
13^1^-keto C=O A_-1A_	1698(−)/1706(+) 1697(−)/1704(+)	8 7	1690(−)/1698(+)	8	1670(−)/1682(+)	12

**Table 3 ijms-25-04839-t003:** Band assignment in (P700^+^–P700) FTIR DS obtained using WT and A_-1_ mutant PSI from *S6803* and *C. reinhardtii* at 293–298 K. Frequency differences upon mutation are indicated by the symbol Δ and are in cm^−1^.

*S6803*					
	WT	N-to-M Mutant		Δ	
13^1^-keto C=O A_-1B_	1697(−)/1710(+)	1683(−)/1691(+)		14/19	
13^1^-keto C=O A_–1A_	1692(−)/1699(+)	1679(−)/1692(+)		13/7	
** *C. reinhardtii* **					
	**WT**	**N-to-K Mutant**	**Δ**	**N-to-D Mutant**	**Δ**
13^1^-keto C=O A_-1B_	1692(−)/1702(+) 1691(−)/1700(+)	1674(−)/1688(+)	18/14 17/12	1673(−)/1688(+)	19/14 18/12
13^1^-keto C=O A_-1A_	1698(−)/1706(+) 1697(−)/1704(+)	1690(−)/1698(+)	8/8 7/6	1670(−)/1682(+)	28/24 17/22

## Data Availability

Data are contained within the article.

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
