# Peer review of "Is the A-1 Pigment in Photosystem I Part of P700? A (P700+–P700) FTIR Difference Spectroscopy Study of A-1 Mutants"

_ijms, 2024, doi:10.3390/ijms25094839_

Round 1
Reviewer 1 Report
Comments and Suggestions for Authors
This paper concerns the nature of the primary donor and acceptor in cyanobacterial Photosystem I. Currently the PA and PB chlorophylls are considered to constitute P700 and the second pair of chlorophylls A0 and A1 are considered ‘accessory’. This work uses FTIR difference spectroscopy to probe the effect of A1 ligand alterations on the formation of P700+A1-. Results are consistent with the idea that A1 pigments are coupled electronically with PA and PB producing a multimeric species that is P700.
This paper deals with 3 mutants, NA600M, NB582M and the double mutants of the same 2 mutations which according to the x-ray crystal structure are adjacent to A1. All 3 mutants show perturbation in P700+ - P700 RT FTIR double difference spectra indicative of perturbation of molecular groups directly impacted by the alteration, or more distant groups that are impacted electrochromically. A close look at these spectra reveal that many of the bands in the FTIR DS display shoulders indicating that several molecular components likely contribute. The authors discuss in detail the different components of the bands leading to the inescapable conclusion that A1 contributes to the multimeric species P700. This conclusion was previously demonstrated for PSI from the green alga C.reinhardtii.
The data was clear and the discussion detailed. I thought the authors communicated in an open style rather than heavily siding one way or the other.
Author Response
Reviewer 1 is happy with the manuscript as is.
Reviewer 2 Report
Comments and Suggestions for Authors
The MS „Is the A-1 Pigment Part of P700? A (P700+ – P700) FTIR Difference Spectroscopy Study of A-1 Mutants.” by Kirpich et al., used FTIR DS to investigated changes to the species P700 in mutant where the ligands to the A1 are altered.
Dear Authors,
The study is very interesting, but the whole MS is not clearly presented and needs to be improved before publication. It requires serious correction from title (there are no periods in the end subtitle or title), abstract, etc. Keywords are missed also. Also, authors should indicate reference for Figure 1. It would be good if the authors introduce number samples and statistical analysis in the MS.
Author Response
Reviewer 2 indicates that he/she is not qualified to assess the English in this manuscript.
Corrections to account for reviewer 2 concerns:
- We modified the legend to figure 1 indicating more clearly where the structure with numbering came from.
- We added the word "Photosystem I" in the title of the manuscript.
- We added six keywords.
We are not sure what to do about the sentence: "It would be good if the authors introduce number samples and statistical analysis in the MS."
Round 2
Reviewer 2 Report
Comments and Suggestions for Authors
After carefully reading MS, I would suggest accepting after the minor revision. Here are some minor comments to help authors improve the paper’s quality.
Comments
Please delete dots in each subtitle.
Please rewrite:
Line 126 -128 „water lig“?
Line 182: „Following on from this work“
Line 190 „Having said that“
Redundant: „see“ in Line 157 -159. (see Figure 1 for Chl a numbering) should be (Figure 1 for Chl a numbering), Line 414. „shown in” Line 358 , “we call” Lines 417, Line 429.
Line 135 -136 „ So useful information about the ET mechanism and Chl participation in it can be deduced from the FTIR DS. – please delete so useful information and change that.
Distances (in Å) should be distance (Å)
Line 142-145 „(P700+ – P700) FTIR DS for WT and mutant PSI samples at RT (293 K) are shown in Figure 3. By subtracting (P700+ – P700) FTIR DS for WT PSI from the corresponding mutant DS, “mutant minus WT” double difference spectra (DDS) are constructed. These DDS are also shown in Figure 3.“ - Please combine in one.
Line 193- 194: „This suggestion is considered in detail below.“ - please rewrite or delete.
Author Response
We have addressed all of the suggested changes:
Line 126 -128 „water lig“? - Added comma.
Line 182: „Following on from this work“ - added the word previous
Line 190 „Having said that“ - Changed to "However"
Redundant: „see“ in Line 157 -159. (see Figure 1 for Chl a numbering) should be (Figure 1 for Chl a numbering), - removed text in parenthesis.
Line 414. „shown in” Line 358 - deleted,
“we call” - deleted
Lines 417, Line 429. - deleted
Line 135 -136 „ So useful information about the ET mechanism and Chl participation in it can be deduced from the FTIR DS. – please delete so useful information and change that. - modified text in this paragraph to make more readable.
Distances (in Å) should be distance (Å) -corrected
Line 142-145 „(P700+ – P700) FTIR DS for WT and mutant PSI samples at RT (293 K) are shown in Figure 3. By subtracting (P700+ – P700) FTIR DS for WT PSI from the corresponding mutant DS, “mutant minus WT” double difference spectra (DDS) are constructed. These DDS are also shown in Figure 3.“ - Please combine in one. - edited accordingly.
Line 193- 194: „This suggestion is considered in detail below.“ - please rewrite or delete. - deleted.